# A Novel Approach for Skin Regeneration by a Potent Bioactive Placental-Loaded Microneedle Patch: Comparative Study of Deer, Goat, and Porcine Placentas

**DOI:** 10.3390/pharmaceutics14061221

**Published:** 2022-06-08

**Authors:** Kritsanaporn Tansathien, Phuvamin Suriyaamporn, Tanasait Ngawhirunpat, Praneet Opanasopit, Worranan Rangsimawong

**Affiliations:** 1Pharmaceutical Development of Green Innovations Group (PDGIG), Faculty of Pharmacy, Silpakorn University, Nakhon Pathom 73000, Thailand; isomeroff@gmail.com (K.T.); puvamin.su.55@ubu.ac.th (P.S.); ngawhirunpat_t@su.ac.th (T.N.); opanasopit_p@su.ac.th (P.O.); 2Division of Pharmaceutical Chemistry and Technology, Faculty of Pharmaceutical Sciences, Ubon Ratchathani University, Ubon Ratchathani 34190, Thailand

**Keywords:** deer placenta, goat placenta, porcine placenta, bioactive extract, skin regeneration, dissolving microneedles

## Abstract

The aims of this study were to investigate the skin regeneration potential of bioactive placenta (deer placenta (DP), goat placenta (GP), and porcine placenta (PP)) and fabricate bioactive extract-loaded dissolving microneedles (DMNs) as a dermal delivery approach. The placentas were water-extracted, and the active compounds were evaluated. Bioactivity studies were performed in dermal fibroblasts and keratinocytes. DMNs were fabricated to deliver the potent bioactive placenta extract into the skin. All placental extracts expressed high amounts of protein, growth factors (EGF, FGF, IGF-1 and TGF-β1), and amino acids. These extracts were not toxic to the skin cells, while the proliferation of fibroblast cells significantly increased in a time-dependent manner. GP extract that exhibited the maximum proliferation, migration, and regeneration effect on fibroblast cells was loaded into DMN patch. The suitable physical properties of DMNs led to increased skin permeation and deposition of bioactive macromolecules. Moreover, GP extract-loaded DMNs showed minimal invasiveness to the skin and were safe for application to human skin. In conclusion, placental extracts act as potent bioactive compounds for skin cells, and the highest bioactive potential of GP-loaded DMNs might be a novel approach to regenerate the skin.

## 1. Introduction

The placenta is an organ containing many active molecules, such as hormones, proteins, lipids, nucleic acids, glycosaminoglycans, amino acids, vitamins, and minerals, which are required for life sustenance and proliferation of the fetus [1]. Placental therapy has been used to stimulate the recovery of disease and tissue regeneration since the early 1900s, in which various clinical applications have exhibited a range of remarkable therapeutic attributes encompassing antioxidant, antimicrobial, anti-inflammation, pain reliever, hair growth stimulation, health improvement, cellular proliferation, tissue regeneration, and wound healing properties [2]. In many countries (e.g., Japan, Korea, and China), porcine placenta (PP) extract containing water-soluble active compounds have been used as a non-prescription drug for analeptic medicine, healthy foods, and cosmetics [3]. Previous studies have shown that peptides from goat placenta (GP) extract and human placenta have numerous bioactivities, such as antioxidant and anti-inflammatory activities [4,5,6]. Deer placenta (DP) also contains some bioactive compounds, such as glycoproteins and alpha-fetoprotein [7]. However, no comparative scientific data of DP, GP, and PP on skin regeneration have yet been reported.

For the healing and repair mechanisms in the skin, growth factors consist of a large group of secreted proteins that are important in regulating and stimulating the growth, proliferation, migration, and differentiation of cells [8]. Placental extracts are rich in growth factors, such as insulin-like growth factor-1 (IGF-1), epidermal growth factor (EGF), fibroblast growth factor (FGF), transforming growth factor-β1 (TGF-β1), vascular endothelial growth factor, granulocyte-colony stimulating factor, granulocyte-macrophage colony stimulating factor, hepatocyte growth factor, and platelet-derived growth factor, referring to physiological effects ranging from immunomodulation, anti-inflammation, wound healing, cellular proliferation, and regeneration [2]. Many exogenous growth factors have been reported as a potential regenerative medicine to replace or repair damaged cells, tissues, and organs [9]. Amino acids are also found in the placentas, including alanine, aspartic acid, arginine, histidine, leucine, lysine, phenylalanine, proline, tyrosine, tryptophan, and valine, which stimulate fibroblast and collagen production and decrease skin pigmentation [2].

The rejuvenating and revitalizing effect of the placenta on the skin have been applied either via mesotherapy or topically along with cosmeceutical aesthetic procedure. Mesotherapy, generally known as “biorejuvenation” or “biorevitalization”, is a technique used to regenerate the skin by using transdermal injection of a multivitamin solution and natural plant extracts to repair the signs of skin aging [10]. However, hypodermic injection is painful, generates dangerous medical wastes, and poses the risk of disease transmission by needle reuse. Self-administered transdermal systems show the improvement of patient compliance and are generally inexpensive [11]. Microneedles can be used to overcome the limitations of conventional approaches. The micron-sized needles arranged on a small patch can deliver hydrophilic macromolecular compounds bypassing the layer of stratum corneum and entering into the skin. This technique provides a fast onset of action, the best patient compliance, self-administration, and improvement of permeability and efficacy. Additionally, dissolving microneedles (DMNs) are fabricated with biodegradable polymers to encapsulate many drugs into the polymer. After inserting these microneedles into the skin, efficient drug delivery without skin irritation has been reported [12]. Therefore, DMNs should be used to deliver bioactive macromolecules into the skin.

The aims of this study were to investigate the skin regeneration potential of bioactive placentals and fabricate a bioactive extract loaded into DMNs as a dermal delivery device. DP, GP, and PP were water-extracted by probe sonication, and the protein, growth factor, and amino acid contents were evaluated. Bioactivity studies were performed for skin cells, such as human dermal fibroblasts (NHFs) and human immortalized keratinocytes (HaCaTs), evaluating the cytotoxicity, cell proliferation and migration, and skin regeneration after UVB-induced skin damage. Subsequently, DMNs were fabricated to deliver bioactive placental extract into the skin. The in vitro skin permeation was determined using Franz-type diffusion cells. Confocal laser scanning microscopy (CLSM) visualized the deposition of macromolecular proteins in the skin. Skin histology was observed after applying the formulation. Moreover, in vivo study was also performed with human volunteers.

## 2. Materials and Methods

### 2.1. Materials

Fresh DP, GP, and PP were gifts from Image Focus Holding Co., Ltd., Bangkok, Thailand, and Payayen Dairy Co., Ltd., Nakhon Ratchasima, Thailand, and CCF Energy Supplement Co., Ltd., Ratchaburi, Thailand, respectively. NHFs and HaCaT cells were obtained from the American-Type Culture Collection (ATCC), Rockville, MD, USA. Dulbecco’s modified Eagle’s medium (DMEM), fetal bovine serum (FBS), trypsin–ethylenediaminetetraacetic acid (EDTA), L-glutamine (Glutamax^TM^), nonessential amino acids, and penicillin–streptomycin were purchased from Gibco BRL, Rockville, MD, USA. The 3-(4,5-dimethyl-2-thiazolyl)-2,5-diphenyl-2H-tetrazolium bromide (MTT) and polydimethylacrylamide (PDMA) were purchased from Sigma, Aldrich, St. Louis, MO, USA. Hyaluronic acid was obtained from the P.C. drug center, Bangkok, Thailand. All other chemical agents were of analytical grade.

### 2.2. Extraction of DP, GP and PP

Fresh DP, GP, and PP were stored in tight containers for freezer storage (−20 °C) before processing. These materials were placed at room temperature for thawing of frozen materials. The extraction method was performed by the sonication method [13]. Then, DP, GP, or PP was mashed with phosphate-buffered saline (PBS), pH 7.4 (1:1), and then weighed 1 g to soak with 20 mL of distilled water. A probe sonicator (Vibra-Cell^TM^, High-Intensity Ultrasonic Processor VCX500, Sonics & Materials, Inc., Newtown, CT, USA) with a frequency of 20 kHz at 20% amplitude was applied for 30 min in an ice bath. Centrifugation was performed at 4000 rpm for 15 min to collect the supernatants. Freeze-drying process was done at −49 °C for 3 days using a freeze dryer (FreeZone 2.5, Labconco, UK) to obtain the dry powder. After that, the extracts were weighed and determined the percentage of yield as Equation (1).
(1)$$\%\mathrm{Yield}=\frac{\mathrm{Weight}\;\mathrm{of}\;\mathrm{extract}}{\mathrm{Weight}\;\mathrm{of}\;\mathrm{fresh}\;\mathrm{placenta}\;\mathrm{used}}\times 100$$

### 2.3. Determination of Protein, Growth Factors and Amino Acids

The soluble protein compositions were determined by SDS-polyacrylamide gel electrophoresis (SDS–PAGE) on a 12% gel. The soluble proteins were dissolved in sterile water (100 mg/mL). The samples were mixed with 2-mercaptoethanol (1:4) to reduce the disulfide bonds and irreversibly denature the proteins. These samples were run at 100 mV for 2 h, and then the gel was stained with a silver staining kit (ProteoSilver^TM^ Silver Stain Kit, Sigma–Aldrich, St. Louis, MO, USA) according to the manufacturer’s protocol.

The total protein content was measured using a bicinchoninic acid (BCA) protein assay kit (Novagen^®^, EMD Millipore Corp., Taunton, MA, USA) using bovine serum protein (BSA) as a standard. Growth factors, such as EGF, FGF, IGF-1, and TGF-β1, were analyzed by the enzyme-linked immunosorbent assay (ELISA kits; Abcam, Cambridge, MA, USA) following the manufacturer’s protocol. The amino acid content was determined by an amino acid analyzer (Hitachi L8900, Hitachi High Technologies Corporation, Tokyo, Japan).

### 2.4. Cytotoxicity Study of Skin Cells

NHF and HaCaT cells were cultured in DMEM supplemented with 10% FBS, 1% penicillin–streptomycin, 1% Glutamax^®^, and 1% nonessential amino acids and incubated under a humidified atmosphere (5% CO_2_, 95% air, 37 °C) until they reached 70–80% confluence. NHFs (10^4^ cells/well) or HaCaTs (5 × 10^3^ cells/well) were seeded into 96-well plates and incubated until cell confluence was achieved. Each extract was diluted to obtain various concentrations (0–5000 µg/mL). After removing the cell medium from the cell plates and washing with PBS, pH 7.4, each concentration of diluted extract was added and incubated for 24 h. The cell viability was measured by the MTT assay [14], in which the diluted extracts were removed, and the cells were washed with PBS, pH 7.4. Medium containing MTT (0.5 mg/mL) was added to the plate and incubated for 3 h. Subsequently, the medium was removed, and 100 µL of DMSO was added to each well to dissolve the formazan crystals that had formed in living cells. Subsequently, the absorbance was analyzed by a microplate reader (VICTOR Nivo^TM^ Multimode Plate Reader, PerkinElmer, Pontyclun, UK) at 550 nm. The percentage of cell viability was calculated using Equation (2).
(2)$$\%\;\mathrm{Cell}\;\mathrm{viability}=\frac{\mathrm{Absorbance}\;\mathrm{of}\;\mathrm{treated}\;\mathrm{cells}}{\mathrm{Absorbance}\;\mathrm{of}\;\mathrm{untreated}\;\mathrm{cells}}\times 100$$

### 2.5. Determination of Cell Proliferation and Migration

NHFs (5 × 10^3^ cells/well) were seeded into a 96-well plate and incubated overnight. Each extract was diluted to obtain various concentrations (0–2000 µg/mL). The old medium was removed, followed by washing with PBS, pH 7.4, and each concentration was added to the cells for 24–72 h. Cell proliferation was determined by the MTT assay, after which the percent cell proliferation was calculated using the same equation as the percent cell viability [14].

For cell migration, an in vitro scratch assay was used to evaluate skin cells [15]. NHF cells (2 × 10^5^ cells/well) were cultured in 6-well plates and incubated until 70–80% cell confluency. Marking lines were performed as a straight line using a 200-μL pipette tip, and then the medium was removed. The extracts (1000 and 2000 µg/mL) were added to NHF cells. The negative control was serum-free DMEM (DMEM supplemented with 1% penicillin/streptomycin), and the positive control was EGF standard solution (200 pg/mL). Images were acquired using an inverted microscope (Nikon^®^ T-DH, Nikon, Tokyo, Japan) after 0, 24, and 48 h of incubation time. The gap of marked lines in the cell plate was measured.

### 2.6. Measurement of Skin Regeneration after UVB-Induced Skin Damage

NHF cells at 8 × 10^3^ cells/mL were seeded into a 96-well plate and incubated at 37 °C and 5% CO_2_/95% humidity. After washing the cells with PBS, pH 7.4, 100 µL of PBS was added. Subsequently, the plate was immediately exposed to UVB rays for 20 min, and then the PBS was removed. Various concentrations of the placental extracts (GP, DP, or PP) were used to treat the cells for 24 h. The percent cell viability was measured by the MTT assay and calculated as Equation (2).

### 2.7. Fabrication and Characterization of DMNs Loading Placenta Extract

Hyaluronic acid (10% *w*/*w*) and PDMA (0.5% *w*/*w*) were used as the main biodegradable polymers for the fabrication of DMNs. Placenta extract or bovine serum albumin-fluorescein isothiocyanate conjugate (BSA-FITC) was mixed with aqueous blends containing biodegradable polymers. All DMNs were fabricated using the micro-molding technique [16]. The formulation was weighted onto laser-engineered silicone micromold templates. The tips of the needle cavities were filled by centrifugation of the microneedle molds at 4000 rpm and 25 °C for 20 min. Microneedle arrays were dried at room temperature for 48 h and then removed from the molds to evaluate the physical properties in terms of appearance, mechanical strength, protein content in DMN patch, and dissolution time.

The physical appearance of microneedles was imaged under a Dino-Lite Edge/5MP digital microscope, AM7915 series (Dino-Lite, Hsinchu, Taiwan), in which the height, width, and interspace of microneedle arrays were measured using dino-capture 2.0 software. For mechanical strength, a texture analyzer (TA. XT plus, Stable Micro Systems, Godalming, UK) with the probe (P/25P) in compression mode was performed at 4.0–20.0 N for 30 s. The percent height change was calculated from Equation (3).
(3)$$\%\mathrm{Height}\;\mathrm{change}=\frac{\mathrm{Height}\;\mathrm{before}\;\mathrm{tested}–\mathrm{Height}\;\mathrm{after}\;\mathrm{tested}}{\mathrm{Height}\;\mathrm{before}\;\mathrm{tested}}\times 100$$

For protein content, DMN patch loading placenta extract was dissolved in PBS pH 7.4 and analyzed protein content as described above. To evaluate the dissolution time of the DMNs in the full thickness of neonatal porcine skin, the skin was stretched on dustless tissue paper and saturated with PBS. DMN patch was pressed into skin with a strength of 10 N of dumble for 2 min. Next, the MNs applied to porcine skin were incubated at 37 °C for different time periods. After withdrawing DMN patch from the skin, the decreased height of DMNs was determined using a Dino-Lite digital microscope.

### 2.8. In Vitro Skin Permeation Study

Abdominal porcine skins were collected from intrapartum stillborn animals from a local farm in Sisaket Province, Thailand. The subcutaneous layer was carefully removed with medical scissors. The skin was 600–700 µm-thick and was kept at −20 °C and thawed in PBS, pH 7.4, at 25 °C before use.

Generally, endogenous proteins and peptides in the skin can interfere with the determined amount of exogenous proteins and growth factors’ permeation into the skin. In this study, BSA-FITC was used as a model macromolecular protein. The skin permeation of macromolecular protein was performed by vertical Franz-type diffusion cells [17]. Approximately 12 mL of PBS, pH 7.4 was added into a receptor compartment and continuously stirred using a magnetic stirrer. The temperature was maintained at 32 °C. The DMN patch was pressed on the skin (2.01 cm^2^ of skin area) with 10 N of dumble for 2 min. A total of 0.5 mL of receiver medium was collected at 1, 2, 4, 6, and 8 h for fluorescence analysis by a microplate reader at an excitation wavelength of 485 nm and an emission wavelength of 535 nm. The receiver compartment was filled with the same volume of PBS to maintain a constant volume. Each sample was analyzed in triplicate.

Fick’s law of diffusion was used as a mathematical model to calculate the parameters of skin permeation. The cumulative permeation amount (μg/cm^2^) against time (h) was plotted, and steady-state flux was measured from the slope of the linear portion in each formulation. The permeability coefficient (K_p_) was the ratio of flux and the donor concentration of the formulation.

### 2.9. CLSM Study

In this study, green fluorescent BSA-FITC was used as a model macromolecular protein [17]. After 8 h of in vitro skin permeation, the treated skins were washed with PBS to remove excess formulation. The fluorescent compound that permeated throughout skin was immersed in sufficient methyl salicylate. The top layer and permeation depth of skin were visualized under a confocal laser scanning microscope (CLSM; an inverted Zeiss LSM 800 microscope, Carl Zeiss, Jena, Germany) equipped with diode lasers (405, 488, and 561 nm). Confocal images were observed at a 10× objective lens. The fluorescence intensities were evaluated at the middle horizontal line of each image using ZEISS ZEN software. The mean fluorescence intensity was plotted against the skin depth.

### 2.10. Skin Histology

The skin was treated with the DMN formulation for 24 h. The skin sample was fixed in 10% formaldehyde solution for 8 h, dehydrated with ethanol, embedded in paraffin wax, and cut vertically along the surface. The cross-sectioned skin was marked with hematoxylin and eosin (H&E) and observed under light microscopy.

### 2.11. In Vivo Human Skin Study

The study involved 15 healthy human volunteers (between 20 and 35 years old) who agreed to participate in a clinical trial. This study was approved by an Investigational Review Board (Human Studies Ethics Committee, Faculty of Pharmacy, Silpakorn University; COE 65.0223-032). Two different points in the forearm were marked, and DMN patches were applied by pressing the thumb on the patch for 30 s. Subsequently, each DMN patch was covered with waterproof tape (OPSITE^◊^ Post-Op, Smith&nephew, Hull, UK) for 24 h. The DermaLab^®^ series (SkinLab Combo; Cortex technology, Hadsund, Denmark) was used to evaluate the skin (transepidermal water loss (TEWL), hydration and erythema).

The TEWL data are presented as the Δ value compared with the baseline (untreated skin). The percent change in skin hydration was calculated (%hydration; Equation (4)). Moreover, the effect of formulations on erythema (vascularity) was determined by generating the value of the percent erythema as shown in Equation (5).
(4)$$\%\;\mathrm{Hydration}=\frac{\mathrm{Ht}\;-\;H}{H}\times 100$$
where Ht is the hydration value of treated skin, and H is the hydration value of untreated skin;
(5)$$\%\;\mathrm{Erythema}=\frac{\mathrm{Et}\;-\;E}{E}\times 100$$
where Et is the erythema value of treated skin, and E is the erythema value of untreated skin [17].

### 2.12. Data Analysis

All data were presented as the mean ± standard deviation (S.D.). Statistically significant difference was analyzed by one-way ANOVA, followed by Tukey’s post-hoc test. In case of in vivo human study, the Wilcoxon signed rank test was determined. The significance level was set at *p* < 0.05.

## 3. Results and Discussion

### 3.1. Physicochemical Properties of DP, GP and PP Extracts

As shown in Figure 1, extraction of DP, GP, and PP provided a brownish-red fibrous texture, in which the percent extraction yields of all extracts ranged from 2.17 to 2. 77. The SDS–PAGE profiles of water-soluble proteins from crude extracts were compared with protein markers (molecular weights ranging from 250 kDa to 10 kDa), in which the bands at 66 kDa, 37 kDa, 24 kDa, 20 kDa, and ~10 kDa were similar to the molecular weights of albumin, glyceraldehyde-3-phosphate dehydrogenase, trypsinogen, trypsin inhibitor, and polypeptide, respectively. All extracts clearly showed bands of high-molecular-weight proteins of albumin and polypeptides (i.e., growth factor); thus, the total protein and growth factors (EGF, FGF, IGF-1, and TGF-β1) showed that the GP extract had higher protein and growth factor contents than the DP and PP extracts (Table 1). 

For the amino acid content, as presented in Table 2, the DP extract was found to contain 14 amino acids in the following order: lysine > alanine > tyrosine > threonine > isoleucine > aspartic acid > glutamic acid > asparagine > valine > serine > leucine > glycine > arginine > histidine. The PP extract had 12 amino acids as follows: cysteine > lysine > valine > glutamic acid > leucine > threonine > phenylalanine > tyrosine > isoleucine > methionine > serine > histidine. The GP extract had nine amino acids as follows: aspartic acid > lysine > methionine > tyrosine > glutamic acid > isoleucine > phenylalanine > arginine > histidine. Although various types of amino acids were found in DP extract, the total amino acid content in GP was higher than that in DP and PP extracts. Aspartic acid was higher content than other amino acids and found predominantly in GP extract. Recently, aspartic acid has been reported as an active ingredient with potential to repair the sign of skin aging and improve the skin condition by stimulating proteins of the dermal extracellular matrix (ECM) [18]. Treatment with amino acids contribute to the collagen and elastin synthesis, leading to increase proteins in fibroblasts. Moreover, the mixture of various amino acids (glycine, proline, lysine, leucine, valine, and alanine) with hyaluronic acid was reported to increased ECM genes in cultured human fibroblasts [19]. Therefore, various types of amino acids in DP extract may also affect the proteins in fibroblasts. 

### 3.2. Cytotoxicity of Placental Extracts on Skin Cells

As shown in Figure 2, none of the placental extracts (GP, DP, and PP) were toxic to fibroblast skin cells at concentrations of 1–2000 µg/mL and significantly exhibited the highest percent NHF cell viability at a concentration of 2000 µg/mL. However, at a concentration of all placental extracts greater than 2000 µg/mL, NHF cell viability was decreased, especially in the PP extract, which decreased the percent cell viability to less than 100%. This result indicated that these placental extracts at concentrations over 2000 µg/mL were toxic to fibroblast cells. For keratinocytes, these extracts at concentrations of 1–5000 µg/mL were not toxic to HaCaT cells and slightly increased the percent cell viability of HaCaTs at concentrations of 1000–2000 µg/mL. These results indicated that at concentrations lower than 2000 μg/mL, the extracts showed no toxicity on either skin cell line and might induce fibroblast proliferation.

### 3.3. Effect of Placental Extracts on Cell Proliferation and Migration

To evaluate the skin-regenerating activity of placental extracts, the effect on NHF cell proliferation was measured. As shown in Figure 3, all placental extracts significantly enhanced cell growth in a concentration- and time-dependent manner (*p* < 0.05). GP and DP extracts at concentrations of 100–2000 µg/mL significantly promoted cell proliferation at 48 h and 72 h when compared to untreated cells, in which the maximum proliferation effect was found at 2000 µg/mL GP and DP extracts. While the PP extract significantly proliferated NHFs at concentrations between 50 and 2000 µg/mL after 72 h of treatment, the highest percent cell proliferation was found with 500 µg/mL PP extract. The results indicated that stimulation of fibroblast proliferation with the GP and DP extracts had a greater effect than that of PP extract.

For the cell migration study shown in Figure 4, treatment with placental extracts significantly accelerated gap area closure compared with a negative control (*p* < 0.05). After treatment with all placental extracts (1000 µg/mL and 2000 µg/mL) for 48 h, the gap areas were completely closed when compared to a negative control (56.17 ± 8.56%). After treatment with placental extracts for 24 h, the width of the scratch area was closed in the following order: GP 2000 µg/mL (88.85 ± 3.42%), GP 1000 µg/mL (87.86 ± 2.46%), EGF (87.81 ± 1.92%), PP 2000 µg/mL (85.65 ± 2.23%), DP 1000 µg/mL (81.14 ± 4.50%), DP 2000 µg/mL (78.75 ± 9.81%), PP 1000 µg/mL (60.04 ± 0.80%), negative control (44.34 ± 8.52%), which indicated that the placental extracts increased human dermal fibroblast migration, and the GP extract provided higher bioactivity on dermal fibroblasts than the other extracts.

Placental tissue is an excellent resource for total protein and growth factors, such as EGF, fibroblast growth factor basic (bFGF), and TGF-β1 [20]. These proteins, growth factors, or cytokines have key roles in dermal or epidermal rejuvenation by inducing proliferation, migration, and extracellular matrix synthesis [21]. Growth factors play an important role in signaling pathways to control cell regeneration and repair. FGF can induce fibroblast skin cell proliferation by accumulating collagen and stimulating angiogenesis [22]. The TGF-β family regulates migration, differentiation, and proliferation. In particular, TGF-β is involved in the proliferative stage, and TGF-β1 is one of three prototypic TGF-β isoforms associated with tissue repair that mediates fibroblast collagen production [23]. EGF also regulates intracellular biochemical pathways such as cell division, cell growth, and cell survival [24]. In this study, all the extracts were found to contain proteins and growth factors (e.g., EGF, IGF-1, TGF-β1, and FGF) as the bioactive compounds to regenerate the skin cells. Therefore, GP extract had the higher protein and growth factor contents, resulting in better bioactivity on dermal fibroblasts than other extracts. Moreover, GP extract containing the highest amino acids that affects the proteins in fibroblasts also resulted in high bioactivity on dermal fibroblasts. 

### 3.4. Skin Regeneration after UVB-Induced Skin Damage

As shown in Figure 5, human dermal fibroblasts were exposed to UVB rays for 20 min, leading to significantly reduced cell viability of 60–70% (*p* < 0.05). After treatment with the placental extracts (GP, DP, and PP), the damaged skins significantly improved the regeneration effect compared to the UVB-irradiated group (*p* < 0.05). The highest cell regeneration was found in the GP extract (2000 µg/mL).

In aged skin, the proliferation, metabolic activity, and functions of skin fibroblasts are impaired, resulting in abnormalities in the synthesis of structural substances such as collagen, elastin, and hyaluronic acid [22]. Growth factors and cytokines such as EGF, vascular endothelial growth factor (VEGF), and TGF-β have been reported to be involved in collagen biosynthesis and to promote skin rejuvenation [25]. In addition, human skin fibroblasts are the primary cells in the dermal layer and are associated with UVB-induced photoaging in the upper layer of these cells. In a previous report, UVB-irradiated human skin fibroblasts significantly decreased the expression of IGF-1, TGF-β1, and EGF [26]. In the previous experiments, these extracts containing some growth factors such as EGF, IGF-1, TGF-β 1, and FGF affected the skin cell proliferation and migration. After the skin cells were damaged by UVB rays, these active ingredients of placental extracts repaired the damaged dermal cells. Consequently, placental extracts (GP, DP, and PP) were useful for skin regenerative activities because they proliferated human dermal fibroblasts by increasing cell regeneration, leading to the induction of fibroblast migration, the upregulation of pro-collagen I, and the repair of UVB irradiation-damaged skin. Moreover, the results showed that GP extract had a stronger bioactive effect on skin cells than DP and PP, resulting from total protein, growth factors, and total amino acids content as the bioactive substances were predominantly found in GP extract. Therefore, GP extract was selected as a bioactive placental extract for delivery into the skin using the microneedling technique.

### 3.5. DMN Loading of Bioactive Macromolecular Compounds

As shown in Figure 6, the physical appearance of hyaluronic acid and PDMA microneedles under the digital microscope displayed sharp conical-shaped microneedle tips (11 × 11 needle arrays) with a height of 557.75 ± 2.11 μm, base width of 299.93 ± 2.31 μm, and interneedle spacing of 600.90 ± 2.22 μm. The percentage of height reduction of DMNs increased progressively with increases in the applied force from 4.0 to 20.0 N. At the average applied force for polymeric microneedles on the skin by human thumbs (20 N), DMNs showed the highest percent height reduction; however, none of the needles were brittle. These MNs deformed when applied to a stainless-steel plate, which is important from a patient safety perspective [27]. 

Protein content in the DMN patch was 461.52 ± 79.43 µg per patch. Moreover, the dissolution of DMNs in the skin exhibited height reductions of 45.47 ± 4.478% and 77.28 ± 0.57% after the application for 2 h and 4 h, respectively, in which the complete dissolution of DMNs was found at an insertion time of 6 h. These results presented an appropriate system of DMNs to deliver bioactive compounds into and throughout the skin.

### 3.6. Skin Permeation and Deposition of Macromolecular Protein-Loaded DMNs

As shown in Figure 7, the cumulative amount of BSA-FITC-loaded DMNs through the skin was higher than that in solution form at every time interval. The flux of DMNs and solution was 0.1745 ± 0.12 µg/cm^2^/h (R^2^ = 0.7521) and 0.0610 ± 0.01 µg/cm^2^/h (R^2^ = 0.9381), respectively, while K_p_ was 1.75 × 10^−6^ cm/h and 0.31 × 10^−6^ cm/h, respectively. This result indicated that DMNs increased the permeability of macromolecular protein through the skin for a 5.65-fold enhancement from solution.

CLSM visualized the skin permeation pathway and evaluated the permeation depth of BSA-FITC-loaded DMNs (Figure 8). The X-Y serial images of skin treated with DMNs exhibited the green fluorescent BSA-FITC between 0 µm and 260 μm skin depth. The spatial distribution of BSA-FITC-loaded DMNs showed the shape of these needle arrays in the skin. For the BSA-FITC intensity, the skin treated with DMNs presented the maximum fluorescence intensity at a depth of 105 µm with a deeper skin permeation depth (260 μm) of green fluorescent BSA-FITC than the skin treated with solution (100 μm). This result indicated greater and deeper deposition in skin of macromolecular protein from DMNs than that from solution.

Bioactive macromolecular proteins and growth factors are hydrophilic compounds and have a limited ability to passively penetrate skin. The outermost layer of skin, the stratum corneum barrier, is a lipophilic layer and allows only small, potent, and moderately lipophilic molecules to partition across it passively into the deeper skin layers [28]. The study showed that DMN patch enhanced the delivery of macromolecular proteins into and through the skin. DMN fabricated from hyaluronic acid can be used to encapsulate peptides and proteins, and the DMN morphology exhibited an acceptable height range for the delivery of drugs into skin with minimal invasion. The DMNs were sufficiently sharp and strong for passage through the layer of stratum corneum, for which the thicknesses of stratum corneum layer, viable epidermis, and dermis are approximately 10–20 µm, 50–100 µm, and 1–2 mm, respectively [29]. After DMN insertion into the skin, possible routes to transport protein and growth factors through the skin was created as micropores into the epidermis layer by using a minimally invasive technique, bypassing the tightly packed stratum corneum barrier and improving the ability to deliver macromolecular protein into the skin [28]. However, DMNs could not be inserted passing through the whole dermal layers completely because of the deformation and elasticity of the skin during the process of penetration [30,31]. Approximately 46% of the needle length was used to create a micropore in the skin, suggesting that the macromolecular proteins remained in the epidermal and dermal layers. Therefore, DMNs exhibited a potential delivery system for biomacromolecules into the active site of action, presenting a suitable dermal delivery system for bioactive placental extracts.

### 3.7. Skin Histology

As presented in Figure 9, the histological images of porcine skin treated with GP extract in solution and the DMN patch indicated a normal appearance of the skin structure, in which the stratum corneum, viable epidermal, and dermal layers remained unchanged. However, DMN provided a minimally invasive approach for the stratum corneum barriers (~130 µm insertion depth), suggesting that GP extract-loaded DMNs were successfully inserted into the epidermal and dermal layers and did not damage the deep skin layer.

In the staining method, the hematoxylin component in the skin stains nuclei of fibroblast cells, rough endoplasmic reticulum, ribosomes, collagen, keratohyalin granules, and elastic fibers as a blue or purple color [32], while the eosin stains cell eosinophilic structures, normally composed of intracellular or extracellular protein (e.g., cytoplasm and connective tissue fibers) as varying shades of pink, orange, and red color [33]. The skin treated with GP extract in the DMN patch showed the blue dots of hematoxylin staining more than the skin treated with GP solution, which the position of blue dots in the skin image suggested that the morphology of fibroblasts and keratinocytes [34]. Therefore, DMNs that can deliver bioactive compounds from GP extract into the skin exhibited the growth of dermal fibroblast and keratinocyte. 

### 3.8. Human Study

After 24 h of DMN application in human volunteers, all subjects were measured for skin changes in TEWL, erythema, and hydration, as presented in Figure 10. No unwanted symptom was observed by visual appearance. TEWL was used to evaluate the puncturing properties at the injection sites of the skin surface, and an increase in TEWL was observed in all DMN arrays with/without GP extract, indicating that the DMN arrays caused disruption of the skin by successfully puncturing the skin and creating permeation pathways [35]. Based on the skin hydration results, almost all skin treated with DMN patches showed an increase in percent skin hydration change, demonstrating a high moisture content of the skin [36]. The erythema value was used to determine the reaction to irritant and allergy. Increasing erythema after applying the DMN patch for 24 h indicated that compression during the application might cause skin redness. Although skin irritation and disruption caused by microneedle arrays were observed, no pain and edema or severe redness of skin was found. 

In this study, the GP extract-loaded DMN patch showed no significant difference from the control DMN patch (no GP extract loaded) after 24 h in the human study, indicating that it served as a suitable dermal delivery device for bioactive macromolecules without altering the microneedle insertion properties. Skin irritation after application of microneedle arrays fabricated from hyaluronic acid was slight and transient, and small transdermal pathways created by these DMNs were also rapidly recovered. These results indicated that the DMN patch fabricated from hyaluronic acid was quite safe [35]. 

Microneedles have been reported as a potent drug delivery system by passing through the stratum corneum permeability barrier, which these needles are also easy to produce and allow the self-administration of drugs without causing pain or bleeding. Low invasiveness of microneedles is important to increase the opportunity for application in population-specific and personalized therapies, especially in pediatrics [37]. Moreover, the development of a minimally invasive needle-sensor to monitor O_2_ levels in the brain using acupuncture needles have been shown that a small tool with minimally invasive has an ability to monitor real-time O_2_ in vivo complex environments with minimizing pain, discomfort, and injury to the patient [38]. In addition, DMNs can be used to deliver immunologically active peptides to the epidermal and intradermal space, which these microneedles array containing 33 × 33 needles with 200 to 125 µm-diameter and 600 µm-height can release the peptides in physiological condition at therapeutic dose [39]. Therefore, DMN patch in this study could be a very useful and effective approach to improve the dermal delivery of bioactive macromolecules, especially potent bioactive placental extracts, without serious damage to the skin.

## 4. Conclusions

Three types of animal placentas (deer, goat, and porcine) were water-extracted and exhibited high protein, growth factor (EGF, FGF, IGF-1, and TGF-β1) and amino acid contents. These placental extracts were not toxic to skin cells at concentrations lower than 2000 µg/mL, while the proliferation of fibroblast cells significantly increased in a time-dependent manner. GP extracts exhibited the maximum proliferation, migration, and regeneration effect on fibroblast cells, suggesting that the GP extract had a stronger bioactive effect on skin cells compared with the other extracts. To overcome the limitation of the skin barrier, DMN patches were successfully fabricated to deliver bioactive macromolecular compounds into and through the skin. Moreover, GP extract-loaded DMNs showed successful skin insertion with minimal skin invasion, supporting their safety for application to human skin. In conclusion, placental extracts act as potent bioactive compounds for skin cells, and the highest bioactive potential of GP-loaded DMNs might play an important role in skin regeneration.

## Figures and Tables

**Figure 1 pharmaceutics-14-01221-f001:**
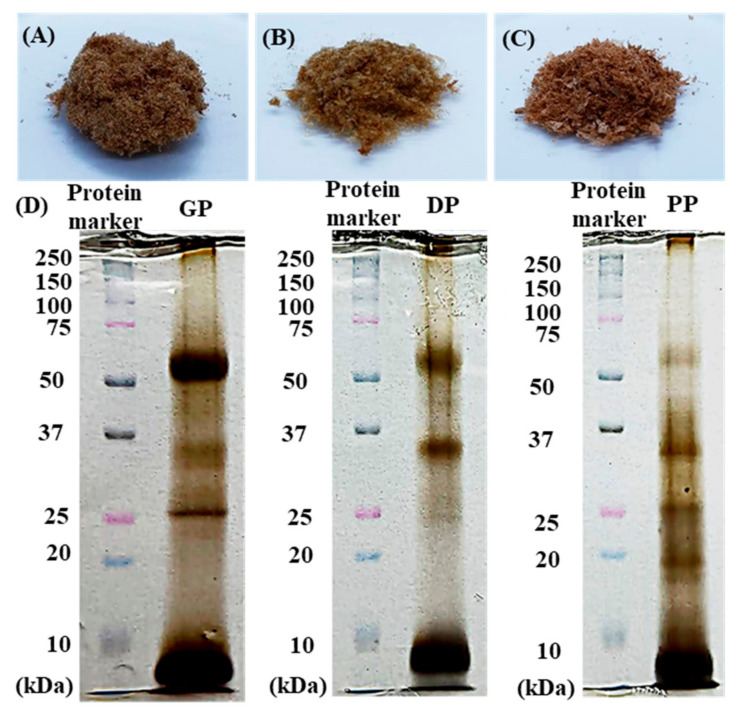
Appearance of GP extract (**A**), DP extract (**B**), and PP extract (**C**) and proteins from the placental extract analyzed by electrophoresis on a 12% SDS–PAGE gel (**D**).

**Figure 2 pharmaceutics-14-01221-f002:**
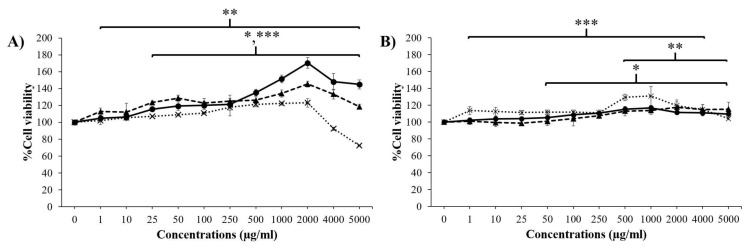
Percentages of cell viability of NHFs (**A**) and HaCaTs (**B**) after treatment with various concentrations of GP extract (
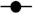
), DP extract (
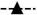
), and PP extract (
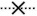
). Data are presented as the mean ± S.D. (*n* = 3). *, **, and *** indicate that the GP extract, DP extract, and PP extract were significantly different from the control group (untreated cells), respectively (*p* < 0.05).

**Figure 3 pharmaceutics-14-01221-f003:**
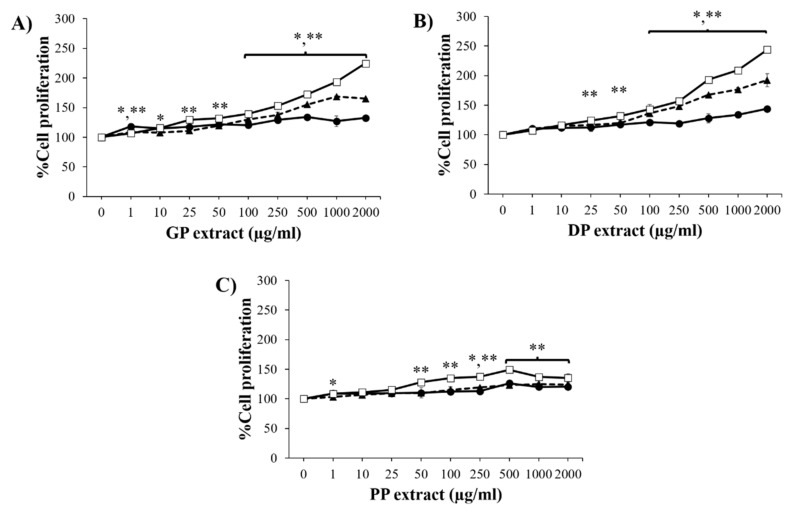
Percent cell proliferation of NHFs after treatment with various concentrations of GP extract (**A**), DP extract (**B**), and PP extract (**C**). Data are presented as the mean ± S.D. (*n* = 3). * and ** indicate that the percent cell proliferation at 48 h (
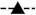
) and 72 h (
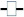
) were significantly different from the untreated cells at 24 h (
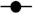
) (*p* < 0.05).

**Figure 4 pharmaceutics-14-01221-f004:**
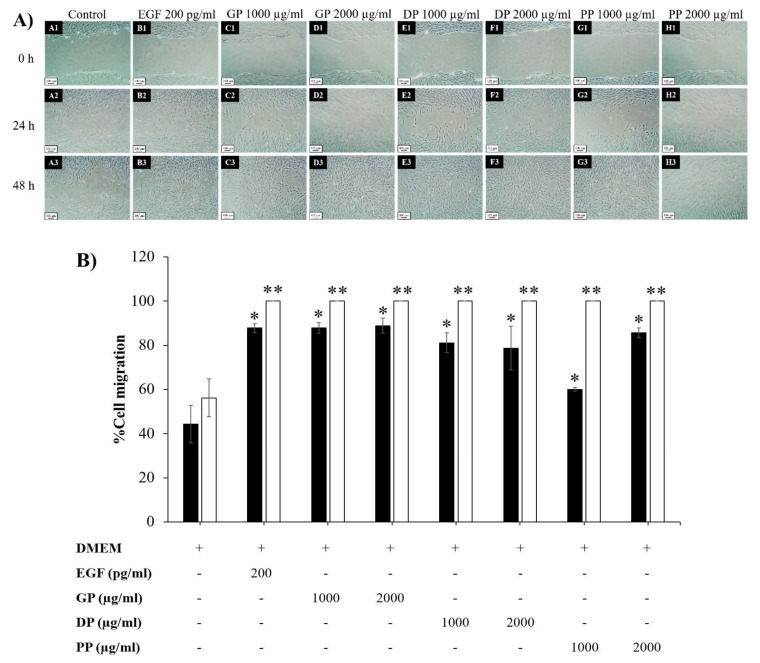
(**A**) Images of NHF cell migration after treatment with 1000 and 2000 µg/mL placental extracts (GP, DP, and PP) compared with a negative control group (untreated cells) and a positive control group (EGF 200 pg/mL) for 0, 24, and 48 h (4× objective lens) and (**B**) width of the scratch area of placental extract (GP, DP, and PP)-treated cells, negative control (untreated cells), and positive control (EGF 200 pg/mL) at 24 h (■), and 48 h (□). * and ** present a significant difference compared with the negative control at 24 and 48 h, respectively (*p* < 0.05).

**Figure 5 pharmaceutics-14-01221-f005:**
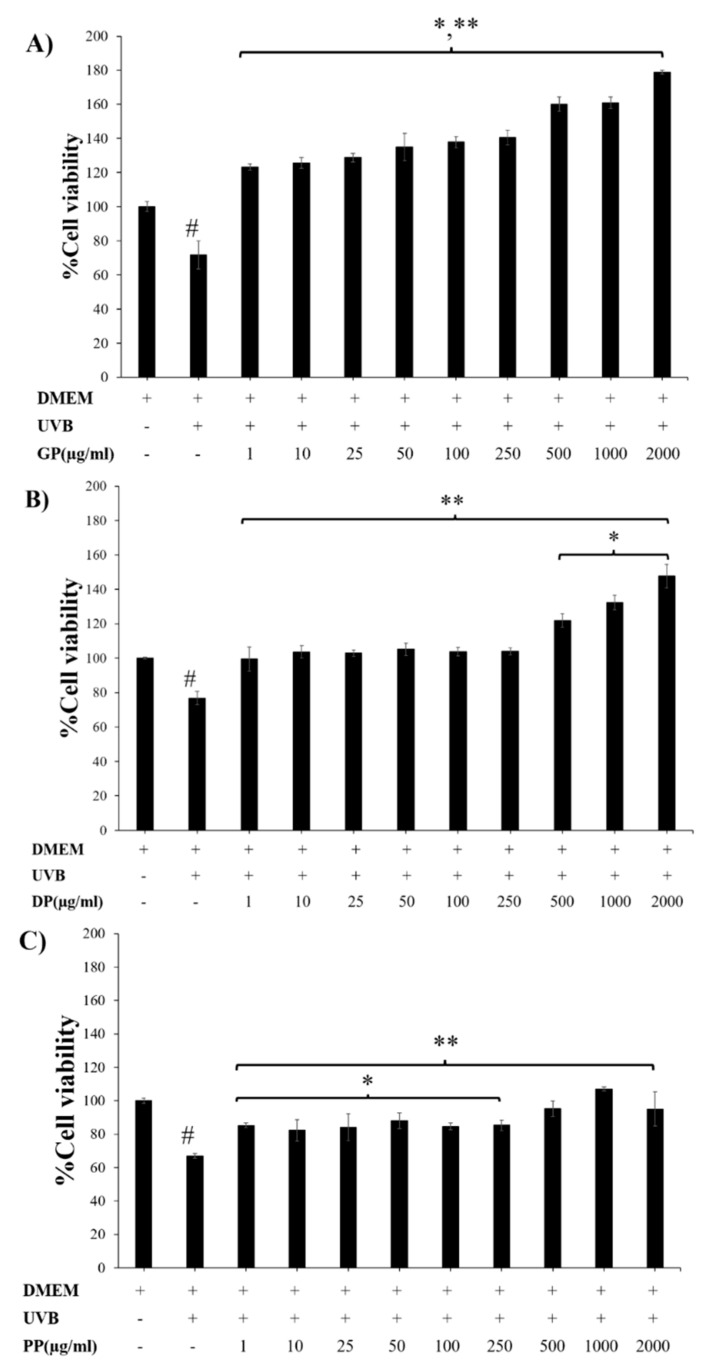
Dermal repair of acute UVB-damaged skin cells by placental extracts GP (**A**), DP (**B**), and PP (**C**) at various concentrations (1–2000 µg/mL) on NHFs treated with UVB ray (*n* = 3). * and ** represent significant differences compared with nontreated UVB and treated UVB (*p* < 0.05), respectively. # denotes a significant difference compared with nontreated UVB (*p* < 0.05).

**Figure 6 pharmaceutics-14-01221-f006:**
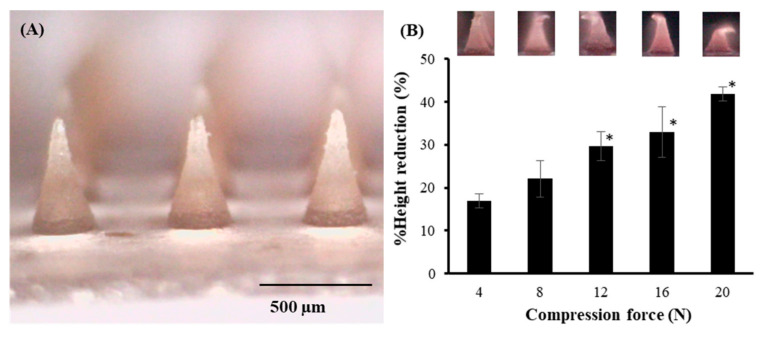
(**A**) Appearance of GP extract loaded DMN arrays and (**B**) the percent height reduction of DMNs after the application of the different compression forces. The data represents the mean ± S.D. (*n* = 3). * represent significant differences from compression force at 4 N (*p* < 0.05).

**Figure 7 pharmaceutics-14-01221-f007:**
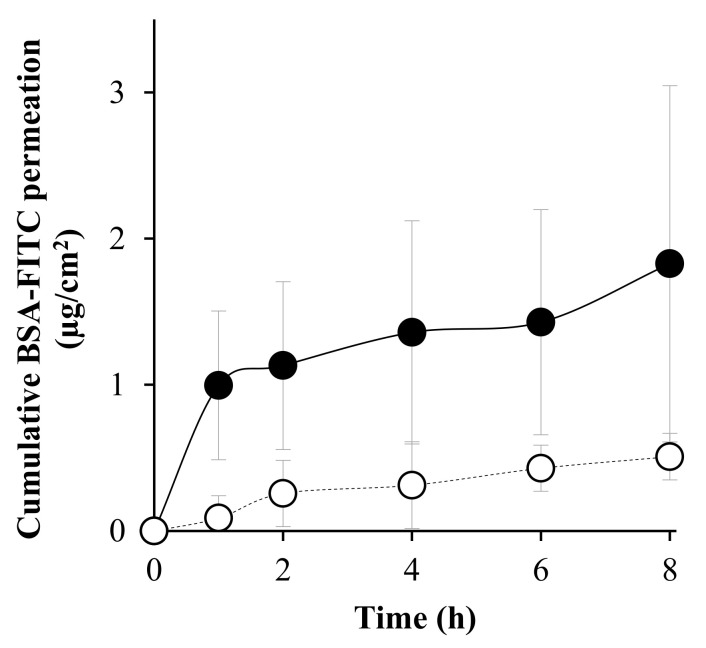
Cumulative BSA-FITC permeation-time profiles of DMNs (●) and BSA-FITC solution (○). The data represents the mean ± S.D. (*n* = 3).

**Figure 8 pharmaceutics-14-01221-f008:**
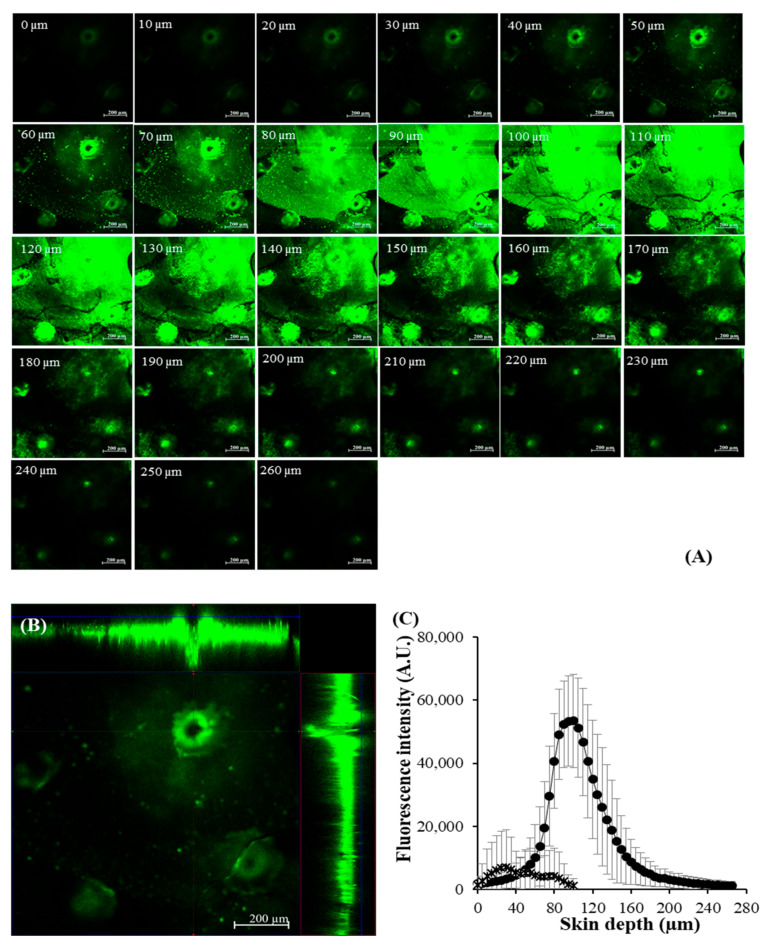
CLSM images of skins treated with DMN loading BSA-FITC for 8 h: (**A**) The X-Y serial images, (**B**) spatial distribution (X−Y, X−Z, and Y−Z planes) image, and (**C**) fluorescence intensity profile at difference skin depths (10 × objective lens).

**Figure 9 pharmaceutics-14-01221-f009:**
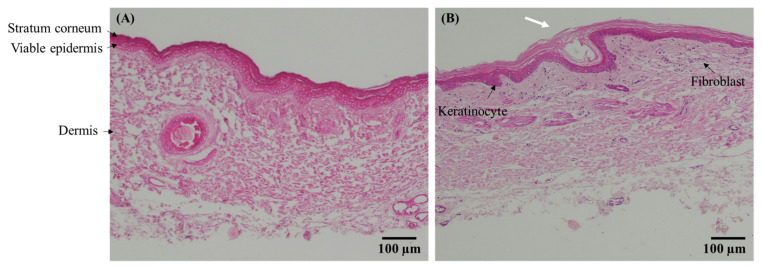
Histology of the skin treated with GP extract in solution (**A**) and DMN patch (**B**) at 24 h (H&E stain, magnification: ×10). The microneedle hole (white arrow) is ~130 µm, reaching the superficial skin.

**Figure 10 pharmaceutics-14-01221-f010:**
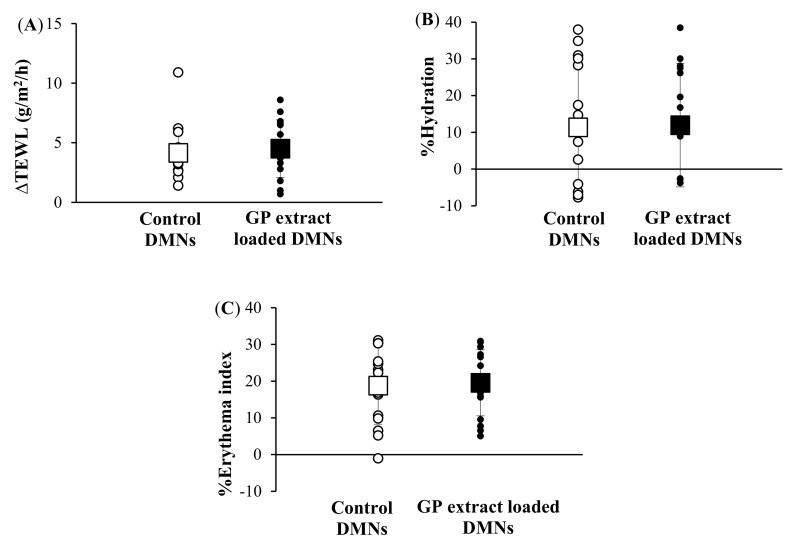
(**A**) ΔTEWL, (**B**) the percent hydration, and (**C**) the percent erythema index after control DMN patch and GP extract loaded DMN patch applied onto the skin of healthy volunteers. The data represents the mean ± S.D. (*n* = 15).

**Table 1 pharmaceutics-14-01221-t001:** The protein and growth factor contents in GP, DP, and PP extract.

Compounds	Contents (per 1 g of Extract)		
	GP Extract	DP Extract	PP Extract
**Protein**	705.51 ± 4.70 mg/g	198.34 ± 3.26 mg/g	151.44 ± 3.65 mg/g
**Growth factors**			
EGF	171.18 ± 8.65 pg/g	39.02 ± 2.92 pg/g	5.79 ± 2.47 pg/g
FGF	401.79 ± 24.07 ng/g	448.99 ± 4.29 ng/g	65.34 ± 6.56 ng/g
IGF-1	155.58 ± 27.82 ng/g	23.07 ± 3.19 ng/g	40.38 ± 6.00 ng/g
TGF-β1	56.52 ± 15.61 pg/g	14.21 ± 0.58 pg/g	55.20 ± 1.21 pg/g

Each data represents mean ± S.D. (*n* = 3).

**Table 2 pharmaceutics-14-01221-t002:** Type and content of amino acids in GP, DP, and PP extract.

Compounds	Contents (per 1 g of Extract)		
	GP Extract	DP Extract	PP Extract
Alanine	ND	2841.08 µg/g	ND
Arginine	0.23 µg/g	34.94 µg/g	ND
Asparagine	ND	1126.98 µg/g	ND
Aspartic acid	23,842.50 µg/g	1449.57 µg/g	ND
Cysteine	ND	ND	124.72 µg/g
Glutamic acid	15.45 µg/g	1356.54 µg/g	48.73 µg/g
Glycine	ND	340.07 µg/g	ND
Histidine	0.08 µg/g	0.38 µg/g	1.04 µg/g
Isoleucine	6.16 µg/g	1608.14 µg/g	16.29 µg/g
Leucine	ND	598.14 µg/g	45.15 µg/g
Lysine	47.69 µg/g	3062.68 µg/g	74.00 µg/g
Methionine	40.14 µg/g	ND	12.24 µg/g
Phenylalanine	3.14 µg/g	ND	26.63 µg/g
Serine	ND	627.39 µg/g	11.35 µg/g
Threonine	ND	1757.01 µg/g	38.64 µg/g
Tyrosine	21.82 µg/g	1860.82 µg/g	16.96 µg/g
Valine	ND	1027.41 µg/g	51.43 µg/g

ND indicated Not Detected.

## Data Availability

Not applicable.

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
