# Peer review of "A Novel Approach for Skin Regeneration by a Potent Bioactive Placental-Loaded Microneedle Patch: Comparative Study of Deer, Goat, and Porcine Placentas"

_pharmaceutics, 2022, doi:10.3390/pharmaceutics14061221_

Round 1

Reviewer 1 Report

- Good to compare the work with other applications such as microneedle in pediatric applications, real-time monitoring and Peptide delivery 

  1. Pires, Liliana R., K. B. Vinayakumar, Maria Turos, Verónica Miguel, and João Gaspar. "A perspective on microneedle-based drug delivery and diagnostics in Paediatrics." Journal of Personalized Medicine 9, no. 4 (2019): 49.
  2. Vieira, Daniela, Francis McEachern, Romina Filippelli, Evan Dimentberg, Edward J. Harvey, and Geraldine Merle. "Microelectrochemical smart needle for real time minimally invasive oximetry." Biosensors 10, no. 11 (2020): 157.
  3. Pires, Liliana R., Isabel R. Amado, and João Gaspar. "Dissolving microneedles for the delivery of peptides–Towards tolerance-inducing vaccines." International Journal of Pharmaceutics 586 (2020): 119590.- 
  • The color code in Figure 1 is difficult to read

Author Response

Point 1: - Good to compare the work with other applications such as microneedle in pediatric applications, real-time monitoring and Peptide delivery 

  1. Pires, Liliana R., K. B. Vinayakumar, Maria Turos, Verónica Miguel, and João Gaspar. "A perspective on microneedle-based drug delivery and diagnostics in Paediatrics." Journal of Personalized Medicine 9, no. 4 (2019): 49.
  2. Vieira, Daniela, Francis McEachern, Romina Filippelli, Evan Dimentberg, Edward J. Harvey, and Geraldine Merle. "Microelectrochemical smart needle for real time minimally invasive oximetry." Biosensors 10, no. 11 (2020): 157.
  3. Pires, Liliana R., Isabel R. Amado, and João Gaspar. "Dissolving microneedles for the delivery of peptides–Towards tolerance-inducing vaccines." International Journal of Pharmaceutics 586 (2020): 119590.- 

Response 1: Thank you for your suggestions. We would like to revise the manuscript in line 493-507, page 14 as following:

Microneedles have been reported as a potent drug delivery system by passing through the stratum corneum permeability barrier, which these needles are also easy to produce and allow the self-administration of drugs without causing pain or bleeding. Low invasiveness of microneedles is important to increase the opportunity for application in population-specific and personalized therapies, especially in pediatrics [37]. Moreover, the development of a minimally invasive needle-sensor to monitor O2 levels in the brain using acupuncture needles have been shown that a small tool with minimally invasive has an ability to monitor real-time O2 in vivo complex environments with minimizing pain, discomfort, and injury to the patient [38]. In addition, DMNs can be used to deliver immunologically active peptides to the epidermal and intradermal space, which these microneedles array containing 33x33 needles with 200 to 125 µm-diameter and 600 µm-height can release the peptides in physiological condition at therapeutic dose [39]. Therefore, DMN patch in this study could be a very useful and effective approach to improve the dermal delivery of bioactive macromolecules, especially potent bioactive placental extracts, without serious damage to the skin.”

 Ref:

  1. Pires, L.R., Vinayakumar, K.B., Turos, M., Miguel, V., Gaspar, J. A Perspective on Microneedle-Based Drug Delivery and Diagnostics in Paediatrics.  Pers. Med. 2019, 9(4), 49. https://doi:10.3390/jpm9040049
  2. Vieira, D., McEachern, F., Filippelli, R., Dimentberg, E., Harvey, E.J., Merle, G. Microelectrochemical Smart Needle for Real Time Minimally Invasive Oximetry. 202010, 157. https://doi.org/10.3390/bios10110157
  3. Pires, L.R., Amado, I.R., Gaspar, J. Dissolving microneedles for the delivery of peptides – towards tolerance-inducing vaccines. J. Pharm. 2020, 586, 119590. https://doi.org/ 10.1016/j.ijpharm.2020.119590

Point 2: The color code in Figure 1 is difficult to read

Response 2: We have already edited as shown in Figure 1

Reviewer 2 Report

Following minor correction required to be along with citation in methodology where ever applicable. 

  1. Line 100: Fresh DP, GP or PP: As it is gifted, any specific storage condition maintained before processing
  2. 105: Freeze drying condition or reference if done previously by someone
  3. Methods 2.3, 2.4, 2.5, 2.6, 2.7, 2.8 and so on. If any of above listed method previously used/described please cite them.
  4. Line 158: MTT assay and calculated as described above. (not data above)
  5. Line 227: waterproof tape? Any specification or local use tape
  6. % Erythema calculated by any instrumentation or manual calculation
  7. Line 505: 2012 two times

Reviewer 3 Report

This is a very detailed and important work, please remind of the following topics--

Normality test was performed for clinical trials? What about non-parametric data? How was the data treated towards non-normal. Figure 1, standard protein markers should be included, or at least add as markers in electrophoresis data. Table 2 GP only shows higher protein value for aspartic acid, please enhance The discussion of the benefits from this protein to skin regeneration and the other proteins found for DP. Until section 3.5 I see the authors fail to discuss the differences between placenta samples based on the protein content. The authors only added a general information about protein at the end of section 3.4. Again, even tough GP had fewer compounds (only aspartic acid as a huge influence) and better cyto-availability, it was chosen but the argument is not convincing. Please expand it more at the end of section 3.5 378 insertion times is not a good term, why not after topical administration. Figure 7, what is the solution ? Pbs? 3.7 does the data fit fickian diffusion? Please indicate the R^2 Section 3.9 there is a difference in histology images and should be included. For instance, there is a sign of stress, elongated . Also some purple dots are presented which I assume are fibroblasts, please carefully check these.

Minor points Grammar lines 37-38, 68-69, Typo at 131
